# Prognosis of Very Elderly Patients after Intensive Care

**DOI:** 10.3390/jcm11040897

**Published:** 2022-02-09

**Authors:** Philippe Michel, Fouad Fadel, Stephan Ehrmann, Gaëtan Plantefève, Bruno Gelée

**Affiliations:** 1Réanimation Médico-Chirurgicale, Centre Hospitalier René Dubos 6 av de l’Ile de France, F95303 Pontoise, France; bruno.gelee@ght-novo.fr; 2Service de Soins Continus, Centre Hospitalier Pierre Le Damany, Rue Kergomar Lannion Cedex, F22303 Lannion, France; fadelf@free.fr; 3CHRU Tours, Médecine Intensive Réanimation, CIC INSERM 1415, CRICS-TriggerSep Network, F37000 Tours, France; stephanehrmann@gmail.com; 4Centre d’étude des Pathologies Respiratoires, INSERM U1100, Université de Tours, F37000 Tours, France; 5Service de Réanimation Polyvalente, Centre Hospitalier Victor Dupouy, 69, Rue du Lieutenant-Colonel Prud’hon, F95100 Argenteuil, France; gaetan.plantefeve@ch-argenteuil.fr

**Keywords:** ICU, very elderly patients, autonomy score, dependency

## Abstract

Elderly patients (over age 85) are increasingly treated in Intensive Care Units (ICU), despite doctors’ reluctance to accept these frail patients. There are only few studies describing the relevance of treatments for this group of patients in ICU. One of these studies defined an age of 85 or over as the essential admittance criterion. Exclusion criteriwere low autonomy before admittance or an inability to answer the phone. Epidemiological data, history, lifestyle, and autonomy (ADL score of six items) were recorded during admission to the ICU and by phone interviews six months later. Eight French ICUs included 239 patients aged over 85. The most common diagnostics were non-cardiogenic lung disease (36%), severe sepsis/septic shock (29%), and acute pulmonary oedem (28%). Twenty-three percent of patients were dependent at the time of their admission. Seventy-one percent of patients were still alive when released from ICU, and 52% were still alive after 6 months. Among the patients which were non-dependent before hospitalization, 17% became dependent. The only prognostic criterifound were the SAPS II score on admission and the place of residence before admission (nursing home or family environment had poor prognosis). Although the prognosis of these elderly patients was good after hospitalization in ICU, it should be noted that the population was carefully selected as having few comorbidities or dependence. No triage critericould be suggested.

## 1. Introduction

Elderly people (aged over 85) represent an increasingly large part of the French population (2.89% in 2014 and 3.11% in 2016 [1]), and the number of elderly patients treated in intensive care units is also increasing as in many other countries [2]. Whether right or wrong, these patients are often considered as frail, having multiple diseases, and being unable to benefit from the intensive care contributions. Even if more and more elderly patients are admitted into our units, there is still great reluctance to accept these patients, although it is unsupported by any evidence. Whether this depends on the units or even the individuals shows that this decision is not based on any rationale [3].

Age is risk factor for intensive care patients [4], but does threshold really exist? This wariness has been reinforced by several studies. One paper [5], in particular, showed catastrophic one-year survival rate (less than 3%) for patients aged over 85 years hospitalized in ICU for hemodynamic failure. Interestingly, the threshold seems to be around the age of 85. This first impression has been confirmed by several recent studies which estimate that patients are even older [6,7].

However, all these studies focus only on survival. Only one recent study [8] estimates the quality of life in one year after ICY stay for patients older than 75. Without any obvious data, many doctors are afraid that these patients will only survive at the cost of significant decrease of their autonomy. This belief is not supported by any studies on patients aged over 75 or 80 years [2,9,10,11]. This age group (75–84) is now well accepted in our units. However, information on elderly patients is missing. For this reason, we conducted study with 6-month follow-up on dependence and survival of the eldest patients.

## 2. Methods

This multicentric, prospective, observational cohort study included all patients aged 85 or older who were referred to French hospitals (one university hospital and seven non-university hospitals) for ICU treatment from 1 January 2015 to 31 December 2016, followed by an admission to ICU up to six months later. The ethics committee (CPP Ile de France 1) issued positive opinion, as did the National Commission for Information Technology and Freedom (CNIL). According to French laws, an information document related to observation studies and requesting its approval had to be handed to each patient or relatives upon admission to ICU. The inclusion criteriused were patients’ age (85 or older) and admission to the ICU. During triage, no department had protocol or recommendation to assist the physician. The decision to admit the patient was the sole responsibility of the ICU physician. Patients can be admitted into ICU from another department, emergency, or directly from their home after support by the mobile emergency service.

The non-inclusion criteriwere bedridden patients already very dependent before admission to the ICU, patients who did not speak French or have neurological disease (dementia, etc.) or psychiatric disease preventing them from being interviewed after 6-month period, patients having no phone number nor contact information, and those having no social security coverage. The exclusion criteriwere the inability to reach the patient or relative after 6-month period. The main key point was the change of autonomy status before intensive care and six months later. Autonomy was assessed by the ADL (Activities of Daily Living [12]) score of six items (Table 1), validated by the geriatrics department, completed with datprovided by the patient or relative. The result is score of zero to six, and an ADL score of less than three was considered as marker of dependence. The secondary key points were survival and any change of residence location. In addition to the ADL score and the residence location before and six months after admission, datcollected were the main antecedents, the Knaus score [13] that describes baseline health status of the patients, the mode of admission to the ICU, the length of hospitalization, the SAPS II score, the main treatments used, the possible existence of withheld or withdrawn life-sustaining therapy decision (WOLST) [14], and death in the ICU. The phone numbers of the patient or relatives, including those of an attending physician or the nursing home, were recorded. Six months later, we used phone interview to get more information about residence location, autonomy, and their overall health status feelings compared to before their intensive care stay.

The optimal sample size was considered as being 157 patients for this purely descriptive study on very large total population, including an error margin of 10% and confidence level of 95%.

Quantitative variables were expressed as the mean (standard deviation) when following Gaussian distribution or median (interquartile range 25–75%) and were compared using the Student T or Mann–Whitney U test, respectively. Factorial analysis was carried out to describe variability. Qualitative variables were expressed as frequencies (95% confidence interval) calculated by angular transformation and were compared using the Chi-square or Fisher’s exact probability test for categorical variables. Datwas tested for normality using the Shapiro-Wilk test. Hazard ratios with 95% CI were used to report the results. The time variable was defined as the number of days from ICU triage. The variables were included in Cox’s regression analysis or logistic regression because all were associated with *p* < 0.2 on univariate analysis. All tests were two-sided with 5% significance and performed using R software [15] (using tidyverse [16] and epiDisplay [17] packages).

## 3. Results

The eight centers enrolled total of 245 patients from 1 January 2015 to 31 December 2016 (Figure 1). Three patients were excluded (dementia). Datafter 6 months could not be collected for another 3 patients. Therefore, the usable sample was reduced to 239 cases.

The baseline clinical characteristics of the study population are shown in Table 2. There were various admission diagnostics, but the most frequent were acute non-cardiogenic respiratory failure (36%), severe sepsis/septic shock (29%), and acute pulmonary oedem (28%). total of 51 (21.4%) patients were only hospitalized in continuing care unit (part of the ICU without invasive ventilation, dialysis, or other heavy techniques). total of 214 patients (89%) were not dependent (ADL Score > 2) before hospitalization in the ICU. total of 68 (28.6%) patients died in the ICU. Death in the ICU was linked to the severity at admission time, but not to age, autonomy before admission, length of hospitalization, nor the treatments used.

## 4. At Six Months

Total of 123 patients (52.3%) were still alive after six months (Figure 2). In addition to the SAPS II score, the predictive factors (for survival) used were the patients origin and their residence location before hospitalization, the worst case being the nursing home and living with family. Only the score SAPS, the dependence before hospitalization, and neurological disease appear to be related to the dependance (Table 3 and Table 4).

As far as the ADL score, only items 1 (personal care) and 5 (continence) changed (Table 5, Figure 3). Among the 88 patients being non-dependent before intensive care and still living after six months, 15 (16.7% (9.9; 26)) of them became dependent. One question dealt with their overall feeling of the change of their quality of life over these six months. One third of the patients answered that they felt better, one third did not notice any difference, and one third felt worse.

Total of 53 patients (22%) went through decision to withhold or withdraw life-sustaining treatment (WOLST). No advance directives have been found for any patient. These patients were not older, not hospitalized more often, nor had more serious medical history. The ADL score showed greater dependence (26.4% versus 14.1%—*p* = 0.003) before the admission for patients for whom WOLST decision had been issued. The mortality rate of patients having undergone procedure to withhold or withdraw life-sustaining therapy was 66% (eight were alive at six months) compared to 17.8% for other patients (*p* < 0.0001). Hospitalization in the ICU was longer. After six months, mortality was 84.9% versus 36.8% (*p* < 0.0001).

## 5. Discussion

The results showed significant early mortality rate (40% of deaths after 30 days), but, after six months, the survival rate was close to 50%. Early mortality in the hospital was similar to what was found in other studies based on similar populations, which confirmed the quality of our sample [8,18,19,20]. More importantly, autonomy was preserved: 60% of the patients having maximum ADL score before hospitalization had the same score after six months, and less than 10% became dependent (ADL score < 3).

The assessment of the autonomy status after six months is not an easy task. The ADL score has been chosen because of its simplicity, which is also its limitation. In order to be able to interview patients by phone, simple score had to be retained. The SF36 score [21], for instance, which is much more complete and well validated, would have been unusable by phone. Therefore, this study is limited to patient autonomy and not quality of life, which is much more complex task to define and approach.

The definition of an ICU varies widely from one country to another, as does the profile of admitted patient types and the seriousness of their status, as well as hospital organization, which makes it hard to assimilate published articles.

Understanding the outcome at 6 months of elderly patients following stay in the ICU could guide the physician in deciding upon the admission of patient to the ICU. If age over 85 is not clear prognostic factor, comorbidities and general condition, as vague as this notion may be, are probably more significant elements. The small size of our sample did not allow the definition of patient groups or simple referral criteria. The residence location before hospitalization was statistically linked to survival and autonomy, but, in reality, it cannot be used: patients living with their family can also be perfectly autonomous or very dependent. Patients living in nursing homes were probably over-selected on admission and, therefore, not particularly representative of this population.

The concept of frailness has recently been introduced in studies focused on patients aged 75 to 85 [22] to define class of non-disabled patients but at high risk of degradation after major hospitalization, (intensive care, surgery, etc.) and on mortality. However, no simple index usable in an emergency department has been validated to date, except the frailness score CSH [23]. However, more than 60% of patients aged 85 and above are classified as “frail” [11,24]. Therefore, these indices are not particularly discriminating for this age group. Criteriusually used (e.g., CFS score [25]) are too complex to collect in an emergency department when deciding to admit to the ICU (walking speed, muscle strength, etc.). Nevertheless, this is an interesting concept which could help refine the triage of patients upon admission.

This study does not show any new criterifor the decision to admit an elderly patient to the ICU. In the service, decisions regarding the continuation of care or the level of commitment in these patients could be based in part on the SAPSII score.

## 6. Conclusions

The positive results in terms of survival rate and autonomy six months after hospitalization in the ICU of patients aged over 85 support an easier admission of these patients. However, these positive results are obtained at the cost of very severe triage, the rules for which still need to be defined.

## Figures and Tables

**Figure 1 jcm-11-00897-f001:**
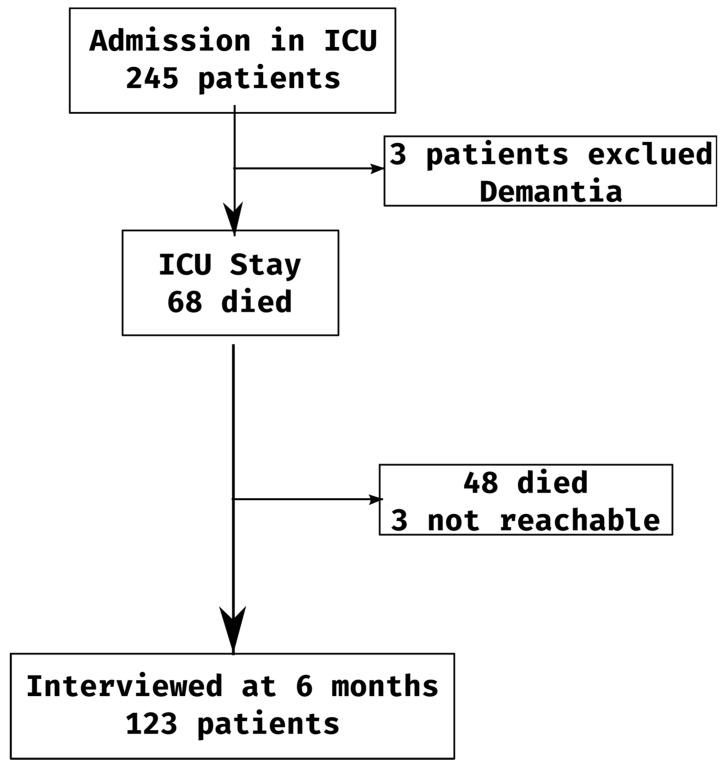
Flow-chart.

**Figure 2 jcm-11-00897-f002:**
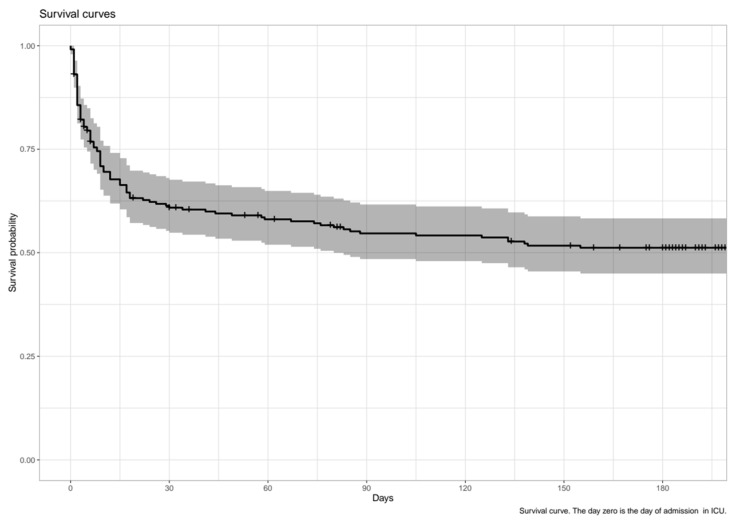
Survival curve {TC “”3”\1 4}.

**Figure 3 jcm-11-00897-f003:**
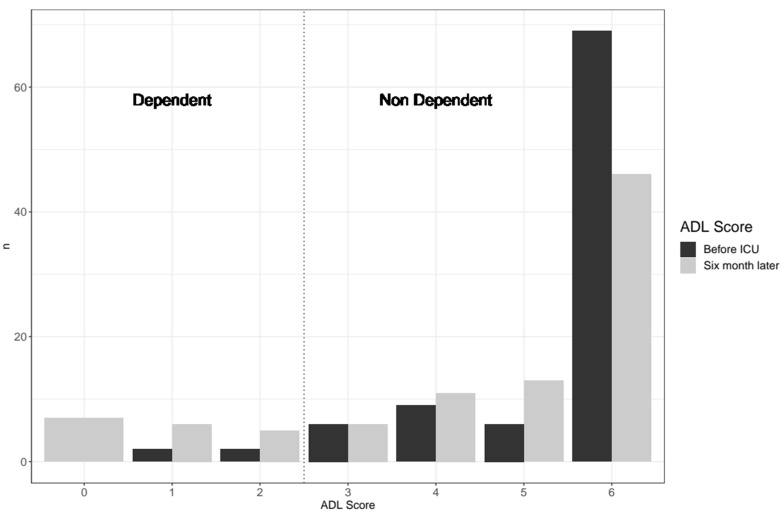
Comparison of the ADL score before the stay in ICU versus six months after.

**Table 1 jcm-11-00897-t001:** ADL Score (Katz and Akpom, [12]).

	Independence: No Supervision, Direction, nor Personal Assistance.	Dependence: With Supervision, Direction, Personal Assistance, or Total Care.
BATHING	Bathes self completely or needs help in bathing only single part of the body, such as the back, genital area, or disabled extremity.	Need help with bathing more than one part of the body, getting in or out of the tub or shower. Requires total bathing.
DRESSING	Get clothes from closets and drawers and puts on clothes and outer garments, complete with fasteners. May have help tying shoes.	Needs help with dressing self or needs to be completely dressed.
TOILETING	Goes to toilet, gets on and off, arranges clothes, cleans genital arewithout help.	Needs help transferring to the toilet, cleaning self or uses bedpan or commode.
TRANSFERRING	Moves in and out of bed or chair unassisted. Mechanical transfer aids are acceptable.	Moves in and out of bed or chair unassisted. Mechanical transfer aids are acceptable.
CONTINENCE	Exercises complete self-control over urination and defecation.	Is partially or totally incontinent of bowel or bladder.
FEEDING	Gets food from plate into mouth without help. Preparation of food may be done by another person.	Needs partial or total help with feeding or requires parenteral feeding.

**Table 2 jcm-11-00897-t002:** Summary of the population before and during ICU stay.

	Mean ± Standard Deviation	IC 95%
	*n*/total (%)	
***n* = 239**		
**Gender** (f)	115/239 (48%)	[42; 55]
**Age**		
Aged 85 years	54/239 (23%)	[17; 28]
Aged 86 years	57/239 (24%)	[18; 30]
Aged 87–88 years	60/239 (25%)	[20; 31]
Aged 89–90 years	35/239 (15%)	[10; 20]
Aged >90 years	33/239 (14%)	[9.6; 19]
**Admission from**		
Already in hospital	91/238 (38%)	[32; 45]
Home vithe mobile emergency service	63/238 (26%)	[21; 32]
Rehabilitation department	10/238 (4.2%)	[1.9; 7.4]
Home vithe emergency department	74/238 (31%)	[25; 37]
**Place of residence**		
Home, alone	107/239 (45%)	[38; 51]
With partner	67/239 (28%)	[22; 34]
With family	28/239 (12%)	[7.8; 16]
Retirement home	21/239 (8.8%)	[5.4; 13]
Nursing home	16/239 (6.7%)	[3.7; 10]
**Medical histories**		
Cancer	27/239 (11%)	[7.4; 16]
Heart failure	60/239 (25%)	[20; 31]
Renal failure	15/239 (6.3%)	[3.4; 9.9]
Respiratory failure	43/239 (18%)	[13; 23]
Neurological disease	23/239 (9.6%)	[6; 14]
Hepatic failure	0/239 (0%)	[0, 0.1]
**ADL score < 3**	25/239 (10.5%)	[6.9; 15.1]
**SAPS II**	53 ± 21	[50; 56]
**Invasive ventilation**	84/239 (35%)	[29; 42]
**Only NIV**	69/239 (29%)	[23; 35]
**Catecholamines**	79/239 (33%)	[27; 39]
**Renal dialysis**	8/239 (3.3%)	[1.3; 6.3]
**Surgery**	19/239 (7.9%)	[4.7; 12]
**Nosocomial infection**	10/239 (4.2%)	[1.9; 7.3]
**Blood transfusion31**	/239 (13%)	[8.8; 18]
**Life-sustaining therapy withheld or withdrawn**	53/239 (22%)	[17; 28]
**Death in the ICU**	68/239 (28%)	[23; 35]
**Length of hospitalization**	5.9 ± 6.1	[5.1; 6.7]

ADL Score: activities of daily living score; SAPS II: Simplified Acute Physiology Score; NIV: Non Invasive Ventilation.

**Table 3 jcm-11-00897-t003:** Dependency six months after ICU stay: risk factors.

	Non-Dependent	Dependent	*p*
** *n* **	76	18	
**Gender**			0.58
F	41/76 (53.9%)	11/18 (61.1%)	
M	35/76 (46.1%)	7/18 (38.9%)	
**Age**	87.6 ± 2.8	87.3 ± 2.2	0.73
**Admission from**			0.55
Already in hospital	21/76 (27.6%)	6/18 (33.3%)	
Home vithe mobile emergency service	22/76 (28.9%)	5/18 (27.8%)	
Rehabilitation department	3/76 (3.9%)	2/18 (11.1%)	
The Emergency department	30/76 (39.5%)	5/18 (27.8%)	
**Place of residence**			0.22
Home, alone	39/76 (51.3%)	8/18 (44.4%)	
With partner	24/76 (31.6%)	5/18 (27.8%)	
With family	4/76 (5.3%)	2/18 (11.1%)	
Retirement home	8/76 (10.5%)	1/18 (5.6%)	
Nursing home	1/76 (1.3%)	2/18 (11.1%)	
**Medical histories**			
Cancer	10/76 (13.2%)	0/18 (0%)	0.23
Heart failure	18/76 (23.7%)	2/18 (11.1%)	0.39
Renal failure	2/76 (2.6%)	0/18 (0%)	1
Respiratory failure	14/76 (18.4%)	4/18 (22.2%)	0.97
Neurological disease	4/76 (5.3%)	6/18 (33.3%)	0.002
**ADL score < 3 before ICU stay**	1/76 (1.3%)	3/18 (16.7%)	0.02
**SAPS II**	46.3 ± 14.9	41.7 ± 11.4	0.16
**Invasive ventilation**	19/76 (25%)	7/18 (38.9%)	0.37
**Only NIV**	29/76 (38.2%)	4/18 (22.2%)	0.2
**Catecholamines**	18/76 (23.7%)	3/18 (16.7%)	0.74
**Surgery**	0.0921 ± 0.291	0.111 ± 0.323	0.81
**Nosocomial infection**	0/76 (0%)	1/18 (5.6%)	0.43
**Renal dialysis**	3/76 (3.9%)	0/18 (0%)	0.91
**Length of hospitalization in ICU**	5.21 ± 3	6.06 ± 4.39	0.33

*n* (%) or mean ± standard deviation; ADL Score: activities of daily living score; SAPS II: Simplified Acute Physiology Score; NIV: Non-Invasive Ventilation {TC “”3”\1 4} {TC “”2”\1 4}.

**Table 4 jcm-11-00897-t004:** Log-binomial regression modeling risk factors for dependence six month later.

	OR	[IC95%]	*p*
**Age**	0.97	[0.70; 1.30]	0.84
**Place of residence before**			
Alone			1
With partner	0.51	[0.08; 2.59]	0.44
With family	0.88	[0.06; 9.04]	0.91
Retirement home	0.00	[NA; NA]	0.99
Nursing home	13.64	[0.58; 483.57]	0.1
**ADL score before admission**			
≥3			1
<3 [dependent]	10,973,843.94	[0.00; NA]	0.99
**SAPS II**	0.95	[0.89; 1.00]	0.09
**Neurological disease**	2.68	[0.21; 29.68]	0.42
**Invasive ventilation**	1.78	[0.23; 13.28]	0.57

ADL Score: Activities of Daily Living score; SAPS II: Simplified Acute Physiology Score {TC “”4”\1 4}.

**Table 5 jcm-11-00897-t005:** ADL score before ICU state and six months later {TC “”2”\1 4}; *n* (%).

	Before Hospitalisation	After 6 Month	*p*
** *n* **	239	94	
Item 1 (Bathing)	185 (77.4)	60 (63.8)	0.017
Item 2 (Dressing)	193 (80.8)	70 (74.5)	0.264
Item 3 (Toileting)	202 (84.5)	75 (79.8)	0.38
Item 4 (Transferring)	204 (85.4)	77 (81.9)	0.54
Item 5 (Continence)	219 (91.6)	77 (81.9)	<0.001
Item 6 (feeding)	220 (92.1)	90 (95.7)	0.34

## Data Availability

Data are available on request from Philippe Michel.

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
