# Peer review of "Prognosis of Very Elderly Patients after Intensive Care"

_jcm, 2022, doi:10.3390/jcm11040897_

Round 1

Reviewer 1 Report

This is an important reference for critical care physicians and for public health. The authors made efforts obviously. There are several questions and suggestions to the authors. 

Only three patients were excluded from the study due to dementia. It is interesting how about the prevalence of dementia since it seemed to be unreasonable. 

The figure should be much more clear or instead of a flow chart. Some wording I cannot understand including limiting care? 

The stratification of age by one year may be not adequate. You know, every one year is of no meaning.

Table 2, short hospitalization, how to define "short".

The medical record history is not complete, such hepatic failure was not included.

The section on results was very confusing. Please explain the result combined with the table.

In table 2 the ADL was also very confusing, it needs more explanation in the footnote.

Figure 2 made a large confusion. What the author wants to present.

Figure 3 please re-plot the figure.

Please reconstruct the whole manuscript, especially the tables, results, and discussion sections.

.

Author Response

Point 1: Only three patients were excluded from the study due to dementia. It is interesting how about the prevalence of dementia since it seemed to be unreasonable. 

Response 1: Dementia was an exclusion criterion. these three cases were therefore inclusion errors. It is impossible to have a number of patients with dementia in the ICUs during the study or proposed in the ICUs because these patients are not admitted to the ICUs even outside of the study.

Point 2: The figure should be much more clear or instead of a flow chart. Some wording I cannot understand including limiting care? 

Response 2: The figure has been replaced by a flow-chart.

The stratification of age by one year may be not adequate. You know, every one year is of no meaning.

Response 3: You are right. But half of the patients are 85 or 86 years old. For example, 5-year-old classes will give one group over 80% and one under 20%. This division is a compromise which seems to me to present the sample better.

Table 2, short hospitalization, how to define "short".

Response 4: The words “short hospitalization” has been changed toAlready in hospital” because they referred to a category of department only French.

The medical record history is not complete, such hepatic failure was not included.

Response 5: No patient with severe liver failure in the age group studied was admitted to our services during the study. The item has been added to table 1.

The section on results was very confusing. Please explain the result combined with the table.

Response 6: The results section has been redone.

In table 2 the ADL was also very confusing, it needs more explanation in the footnote.

Response 7: I added a table detailing the ALD score in the methods

Figure 2 made a large confusion. What the author wants to present.

Response 8: The figure has been redone.

Figure 3 please re-plot the figure.

Response 9: The figure has been redone.

Please reconstruct the whole manuscript, especially the tables, results, and discussion sections.

Response 10: The discussion section, table etc. havent been redone.

Reviewer 2 Report

What kind of triage criteria do you propose in the end?
you can also make a comparison criterion with other countries, not necessarily Turkey

Author Response

Point 1:What kind of triage criteria do you propose in the end?

Response 1: I edited the end of the thread. I propose in particular the SAPS II score, calculated routinely in all ICUs, to assess the level of commitment or the continuation of care but not specifically validated in this age group.

Point 2: you can also make a comparison criterion with other countries, not necessarily Turkey

Response 2: I deleted the Turkish ICU example. A comparison between countries would be interesting but complex, the definition of an UCI varying a lot from one system to another.